# Estimating the Economic Value of Change in Ecosystem Habitat Quality in South Korea Using an Integrated Environmental and Economic Analysis

Hyun No Kim * and Hwanhee Ryu

Sustainable Strategy Research Group, Korea Environment Institute, Bldg B, 370 Sicheong-daero, Sejong-si 30147, Republic of Korea
* Correspondence: hnkim@kei.re.kr; Tel.: +82-44-415-7985

**Abstract:** Jeju Island is one of the most prominent ecotourism destinations in South Korea and has steadily been the subject of land-use development. In this study, the economic value and benefits of habitat quality changes on Jeju Island were estimated through an integrated environmental and economic analysis linking the unit values of ecosystem services to the physical habitat quality measures predicted by the InVEST Habitat Quality Model. A choice experiment survey was conducted to estimate the willingness-to-pay for the changes in habitat quality at 12 tourist sites using a hybrid econometric model. The results indicate the presence of heterogeneity in respondents' preferences for ecotourism destinations. Visitors were most sensitive to changes in the habitat quality level among three attributes: on-site facilities, information and interpretation services, and habitat quality. Based on the willingness-to-pay for each tourist site, the total benefit resulting from the improvement of habitat quality in the 12 tourist sites is substantial. The integrated environmental and economic analysis framework used in this study would effectively produce plausible economic values generated by environmental goods and services. The findings also provide a basis for considering the importance of economic benefits provided by ecosystem services in decision-making.

**Keywords:** non-market valuation; habitat quality; integrated environmental and economic analysis; choice experiment; hybrid econometric model; ecosystem services

## 1. Introduction

Global ecosystems have changed rapidly and extensively over the past five decades as humanity has constantly interrupted natural ecosystems by introducing artificial forms of land development [1]. While South Korea has achieved rapid economic growth over the past few decades, extensive and intensive land development has damaged wildlife habitats and reduced biodiversity. The decline in biodiversity is expected to continue as development activities have increased steadily over the past 20 years (2002–2021) [2]. However, awareness of ecosystems' benefits for humanity has also been increasing. There is a growing need to evaluate the current situation objectively and scientifically to establish a foundation responsible for national policy-making processes for a reliable supply of ecosystem goods and services. To this end, it is first necessary to understand the value of ecosystem services.

In response to this, studies have been conducted to measure the economic value of environmental conservation. Wassihun et al. [3], Bhat and Sofi [4], Platania and Rizzo [5], Robles-Zavala and Chang Reynoso [6], Sardana [7], Tonin [8], and Iranah et al. [9] estimated tourists' or/and residents' willingness-to-pay (WTP) for the conservation of ecosystem services in various ecotourism destinations using the contingent valuation method (CVM). Obeng et al. [10], Hynes et al. [11], Dushani et al. [12], Hassan et al. [13], Tan et al. [14], Khan et al. [15], Kularatne et al. [16], and Owuor et al. [17] applied a choice experiment(CE) to measure respondents' preferences for the improvement of environment and identify

the factors that affect their WTPs. These studies have estimated the economic value of ecosystem services by using a CE and including biodiversity as a critical attribute of ecotourism. Based on previous studies, our study applied a CE with ecotourism as the mediator to estimate the economic value of changes in the habitat quality in the main tourist destinations on Jeju Island. However, this study is different from the previous studies in that it utilizes the pivot-style experimental design approach allowing respondents to reveal their preferences more effectively [18]. In addition, this study applies advanced hybrid econometric models to estimate more accurate WTP in empirical aspects. Finally, although numerous studies have been conducted to estimate the value of ecosystems, studies on the economic valuation of habitat quality are very rare. Our study differs from existing studies in that it estimates the economic/social benefits of changes in habitat quality in tourist destinations throughout Jeju Island.

In this study, we estimate the economic value of changes in habitat quality, which we refer to as an ecosystem's ability to provide appropriate conditions for wildlife habitation and an indirect indicator to evaluate the level of biodiversity [19]. Therefore, habitat quality corresponds to supporting services essential to enable other services among the various classifications of ecosystem services. According to Brooks [20], habitat quality is assessed by the Habitat Suitability Index on a scale of 0.00 to 1.00, where sites of excellent habitat quality receive high scores (e.g., 0.70–1.00), and sites of poor habitat quality receive low scores (e.g., 0.00–0.30).

This study estimated the economic value and benefits of changes in ecosystem habitat quality using an integrated environmental and economic analysis framework. It was necessary to link the quantification of physical environmental impacts and the monetization of the environmental impacts, to apply the integrated environmental and economic analysis framework. Figure 1 illustrates the economic value and benefit estimation procedures for change in habitat quality after applying the integrated analysis framework. Kim et al. [21] evaluated habitat quality according to changes in land cover in 17 cities and provinces in South Korea using the InVEST habitat quality model. This study used the Jeju ecotourism survey conducted in 2019 to monetize the environmental impact. The survey was conducted in 2019 to identify public awareness and preferences for ecotourism destinations on Jeju. The survey items included estimates of the economic value of changes in habitat quality in the 12 main ecotourism destinations on Jeju by applying a choice experiment (CE). To evaluate the changes in habitat quality in each of the 12 main tourist destinations, we used information on areas with low habitat quality (0.00), medium habitat quality (0.50), and high habitat quality (1.00) based on the physical impact assessment conducted by Kim et al. [21]. After establishing a suitable econometric model using the survey data applying a CE, we estimated the willingness-to-pay (WTP) for habitat quality change in each tourist destination. Finally, we estimated the social benefits using the results of the physical impact assessment (habitat quality change due to changes in land cover) and the WTP obtained.

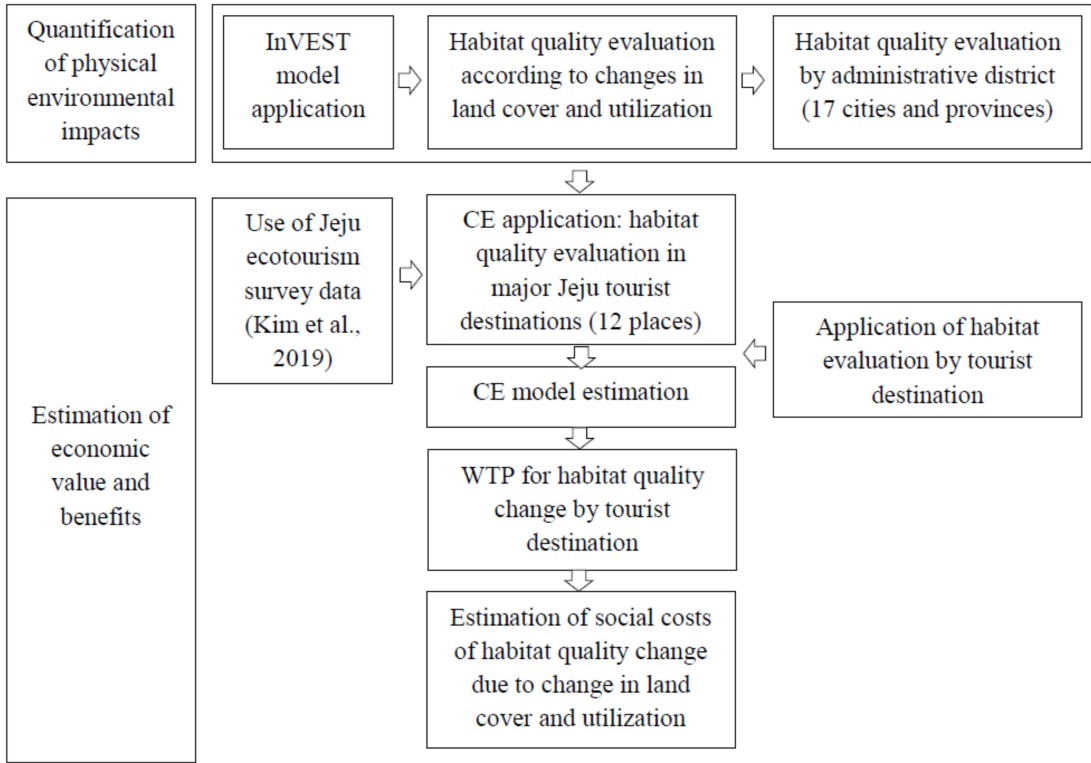

**Figure 1.** Illustration of Integrated Environmental and Economic Analysis to estimate the economic value and benefits of changes in ecosystem habitat quality.

## 2. Materials and Methods

### 2.1. Survey Outline

This study provides policy data to promote sustainable ecotourism by identifying national awareness and preferences for ecotourism destinations on Jeju. An overview of the questionnaire design is presented in Figure 2. The survey was conducted on adults aged 19–59 who had visited the 12 main tourist destinations on Jeju over the past 5 years as of September 2019. An online survey was conducted, and the samples were extracted using quota sampling by gender, region, and age. The maximum margin of sampling error at the 95% confidence level was ±2.82%. A preliminary survey was conducted for about two weeks to refine the questionnaire's items and content. The primary survey was conducted from 21 October to 25 October 2019, and there were 3000 samples. The survey comprised four parts: Jeju travel behavior, Jeju ecotourism usage behavior, an evaluation of ecotourism service improvement, and respondents' demographic characteristics.

The primary survey was conducted on respondents who visited Jeju for tourism between September 2014 and September 2019 (5 years). The section on Jeju travel behavior included questions about inconveniences when traveling on Jeju, details of travel activities, modes of transport, and type of visit. The section on Jeju ecotourism usage behavior asked about visiting frequency, travel itineraries, travel expenses, accommodation, satisfaction with tourist destinations, and revisit intention. The section evaluating the quality of ecotourism services asked about the respondents' most preferred tourist destination and the date and time of their visit, after which the respondents were asked to evaluate attributes related to the CE about their most preferred destination. After evaluating the attributes related to a CE, a CE scenario was presented to the respondents. The last part of the survey comprised questions about the respondents' demographic characteristics, such as occupation and income. They were then asked to rate the suitability and their understanding of the information provided for the survey.

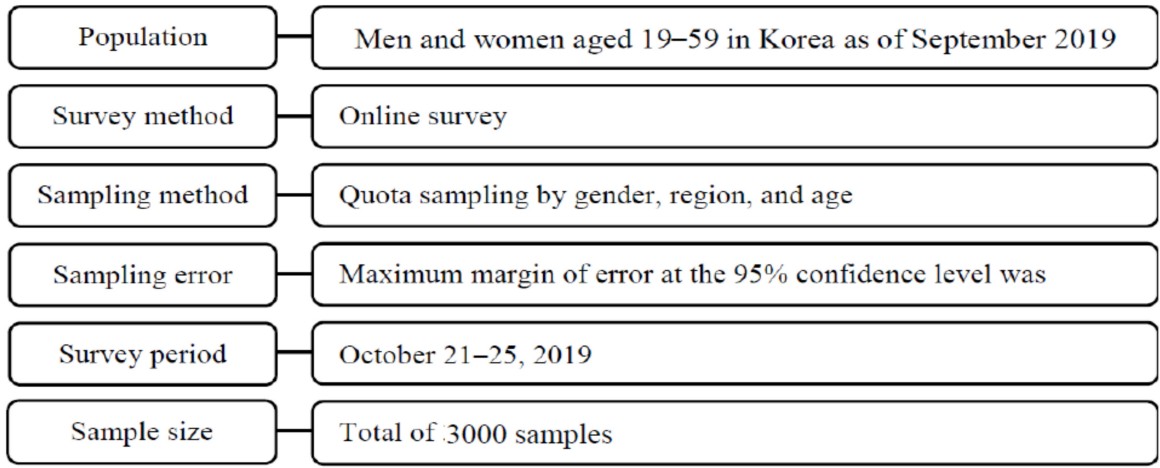

**Figure 2.** Outline of the survey design.

*2.2. Choice Experiment Design*

A CE is often used for the economic valuation of non-market goods or services to estimate WTP for the change in the level of individual properties of the subject being analyzed (goods/services). An economic valuation that applies a CE is conducted according to systematic procedures, as illustrated in Figure 3 [22].

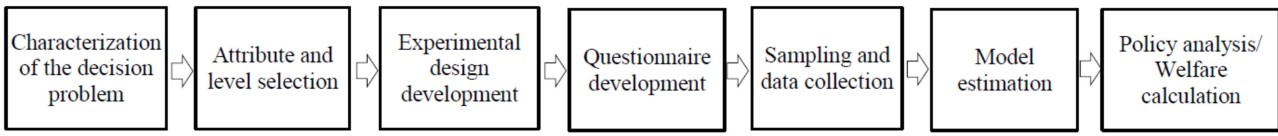

**Figure 3.** Source: Champ et al. [22]. CE design and valuation procedures.

After determining the study's analysis subject and decision problem, the subject's key attributes and the level of individual attributes were selected. As mentioned in the previous section, this study selected the critical attributes of ecotourism and determined the level of individual attributes to estimate the economic value of changes in the level of ecological habitat quality. Four key attributes related to ecotourism were selected for this study, and the individual attributes were defined as follows:

- Amenities in the tourist destination: The amenities used in the tourist destination included restrooms, parking lot, accommodation, information center, convenience of public transportation, and parking lot accessibility.
- Information/interpretation services in the tourist destination: The level of services provided in the tourist destination included signboards, signposts, tourist brochures, interpretation guides, and signs describing the plants in the tourist destination or park.
- Habitat quality in the tourist destination: We considered habitat quality an indirect index to evaluate biodiversity. Habitat refers to a space inhabited by wildlife, and habitat quality refers to the ability of a place to provide appropriate living conditions for the wildlife. Habitat quality was rated on a scale of 0.00–1.00, with a value closer to 1.00 indicating better habitat quality.
- Tourist destinations admission fees: Admission fees were per adult in each tourist destination.

In the literature, most environmental valuation studies applied hypothetical taxes or charges as payment methods to examine changes in the analysis subject (environmental quality). Unlike previous studies, this study used the actual admission fee per adult for each tourist destination rather than a hypothetical payment method as a trade-off payment tool for the three attributes (on-site facilities, information and interpretation services, and

habitat quality). Admission fees of the 12 major tourist destinations on Jeju are illustrated in Figure 4. Table 1 contains the levels of measurement and the changes in the individual attributes. This study promoted respondents' understanding of the definition of habitat quality in a tourist destination and the level of each attribute to a CE and provided the information summarized in Figure 5 to perform a more accurate habitat quality evaluation.

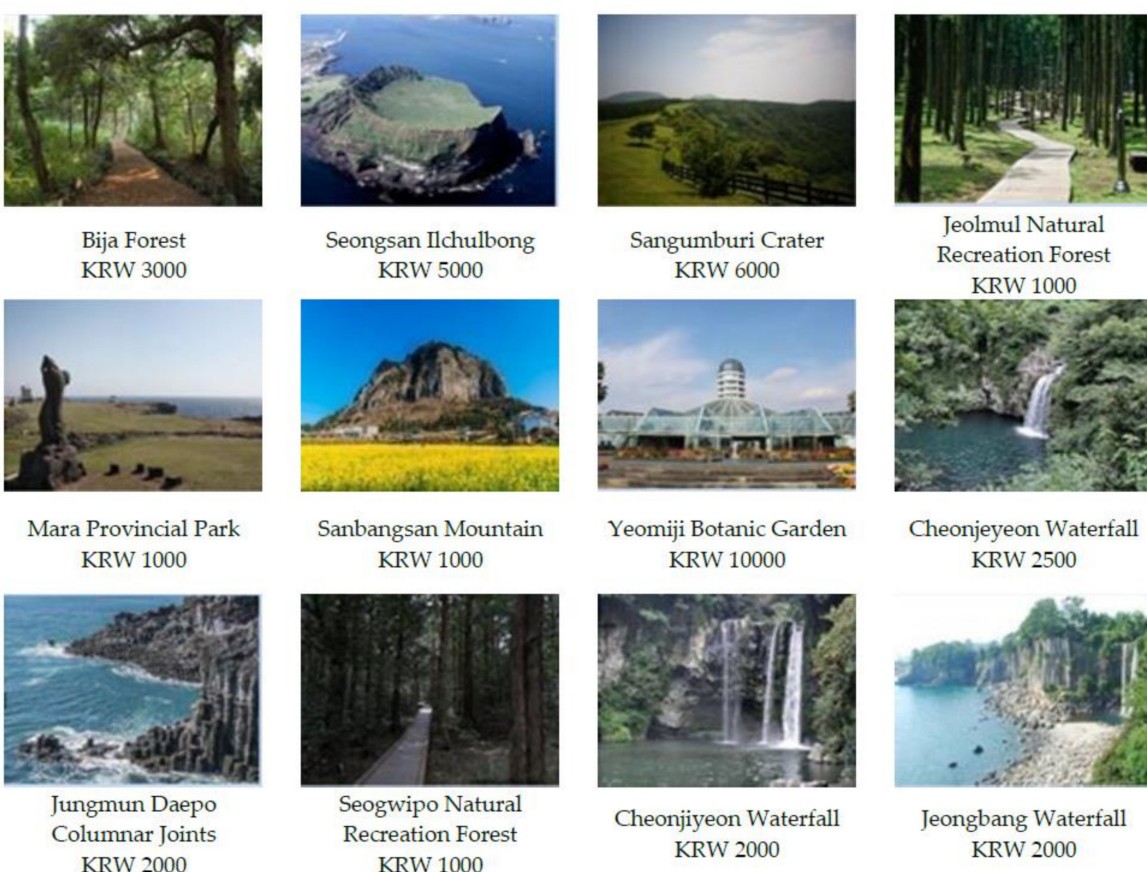

**Figure 4.** Admission fees for 12 tourist destinations on Jeju.

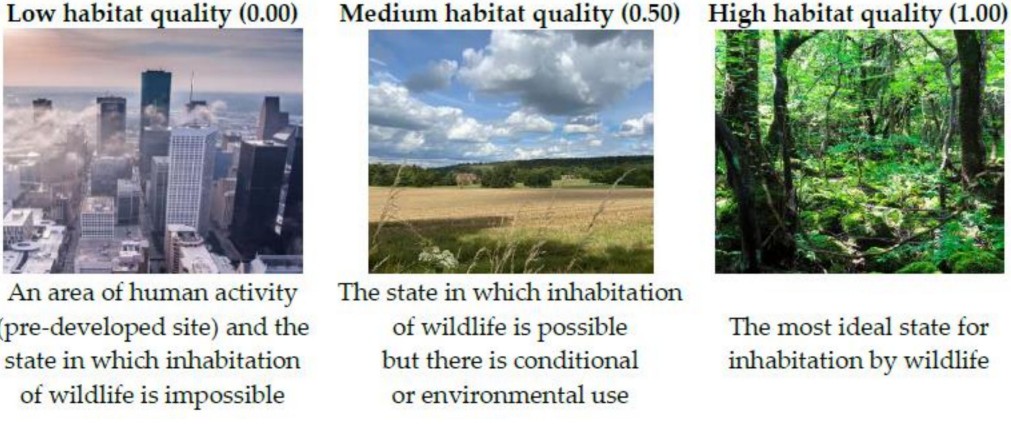

**Figure 5.** Examples of habitat quality evaluation.

After determining the level of the key and individual attributes, the choice sets given to the respondents were determined by applying the experimental design. The number of choice alternatives and the number of choice sets that were included in the individual choice sets were determined through the experimental design. This study applied a

pivot-style experimental design, which is frequently used for experimental designs. First, the respondents were asked to select their most preferred destination among 12 tourist destinations on Jeju and evaluate the relevant attributes of the tourist destinations based on their own experiences. The level of the individual attributes they evaluated was presented in one of the choice alternatives of tourist destinations given in the CE. The attribute levels of the remaining choice alternatives (hypothetical alternatives) were designed by applying the level of change presented in Table 1 based on the respondents' evaluation alternatives. Individual choice sets provided a total of three selectable alternatives, which included respondents' evaluation alternatives. A set of choice alternatives consisting of a combination of different attribute levels for each respondent was repeatedly presented, and the respondents performed six CEs. Table 2 illustrates an example of the choice alternatives given to the respondents by applying the pivot-style design.

**Table 1.** Attributes related to ecotourism, measurement level, and change level.

| Attribute | Measurement Level | Change Level |
| --- | --- | --- |
| Satisfaction with amenities in the tourist destination | 0–100 points | 30% decrease, 10% decrease, no change, 10% increase, 30% increase |
| Information, interpretation, and guide services in the tourist destination | 5-point scale | 1 level decrease, no change, 1 level increase |
| Habitat quality in the tourist destination | 0.00–1.00 | 50% decrease, 25% decrease, no change, 25% increase, 50% increase |
| Admission fee for the tourist destination | Admission fee per adult | 30% decrease, 15% decrease, no change, 15% increase, 30% increase |

**Table 2.** Examples of choice alternatives applying a pivot design.

| Attributes | Alternative 1 (Tourist Destinations Evaluated by Respondents) | Alternative 2 (Hypothetical Tourist Destination 1) | Alternative 3 (Hypothetical Tourist Destination 2) |
| --- | --- | --- | --- |
| Satisfaction with amenities in the tourist destination | Response value obtained | 10% decrease | 30% increase |
| Information, interpretation, and guide services in the tourist destination | Response value obtained | No change (value obtained) | No change |
| Biodiversity quality level | Response value obtained | 50% decrease (value obtained) | 50% decrease |
| Admission fees at the tourist destination (per adult) | Fixed value | 30% decrease | 15% decrease |
| Choose your most preferred tourist destination. | ☐ | ☐ | ☐ |

### 2.3. Empirical Model

CE data are generally analyzed through conditional logit (CL), and as illustrated in Table 2, three choice alternatives ($j$ = 1,2,3) were given to the individual respondents ($i$). When a CE is repeated six times ($n$ = 1, . . . , 6), the conditional indirect utility function can be expressed as follows:

$$
\begin{aligned}
U_{1,n}^i &= V_1^i + \epsilon_{1,n}^i \\
U_{2,n}^i &= V_{2,n}^i + \epsilon_{2,n}^i \\
U_{3,n}^i &= V_{3,n}^i + \epsilon_{3,n}^i
\end{aligned}
\tag{1}
$$

The conditional indirect utility function in Equation (1) was divided into the part observable by the researcher ($V$) and the part not observable ($\epsilon$). As illustrated in Table 2,

the observable part of the choice alternative experiments evaluated by the respondents (alternative 1) among the choice alternatives applying the pivot design is indicated as $V_1^i$ instead of $V_{1,n}^i$ as the values are the same during the iterations of a CE *n* times.

According to Champ et al. [11], the CL model is based on several assumptions to simplify the quantitative analysis. One of the assumptions is that all the respondents have the same preference structure. In other words, the estimation coefficient of individual attributes is limited to the same for all the respondents. The following econometric models can reflect the heterogeneity of respondents' preferences: (1) a model that intersects the demographic variables of the respondents (e.g., age and income) and attribute variables and includes them as explanatory variables, (2) a latent class model, and (3) random parameter logit (RPL) model.

The RPL model has several advantages in that it can consider the heterogeneity of preferences and identify the ratio of respondents with positive and negative preferences for the individual attributes of the choice alternatives. In this study, we estimated the basic CL and RPL models. Moreover, when applying the pivot style, the level of individual attributes (values) of the standard alternatives (tourist destinations evaluated by the respondents) was the same during the n iterations of the experiment. Thus, respondents could handle the other two hypothetical alternatives (hypothetical tourist destinations 1 and 2) differently from the standard alternatives. In other words, there could be a high correlation among hypothetical alternatives among the three alternatives, which may violate the basic assumption of the error terms such as "independent and identically distributed." This problem can be solved by including additional error components (EC) in utility functions. Therefore, this study additionally estimated the model, including the EC in the RPL model. The conditional indirect utility function is as illustrated in Equation (2):

$$
\begin{aligned}
U_{1,n}^i &= V_1^i + \sigma_1 \varnothing_{1,n}^i + \epsilon_{1,n}^i \\
U_{2,n}^i &= V_{2,n}^i + \sigma_2 \varnothing_{2,n}^i + \epsilon_{2,n}^i \\
U_{1,n}^i &= V_{3,n}^i + \sigma_3 \varnothing_{3,n}^i + \epsilon_{3,n}^i
\end{aligned}
\tag{2}
$$

The observable part (*V*) of Equation (2) comprised $\beta^i X_{j,n}^i$ and illustrates the heterogeneity of the respondents (*i*) in individual attributes, and $\sigma_j (j = 1, 2, 3)$ indicates an estimated coefficient representing the variance in individual alternatives, which refers to the additional EC (standard deviation, SD) previously mentioned. Here, $\varnothing_{j,n}^i$ is presumed to follow a standard normal distribution ($\varnothing_{j,n}^i \sim N[0,1]$).

## 3. Results

### 3.1. Estimation Results

This study estimated three different econometric models: a CL model—a basic model that did not consider the heterogeneity of preferences; an RPL model reflecting the heterogeneity of preferences; and the hybrid model, which combines RPL and EC allowing the heteroscedasticity among choice alternatives in the RPL model (RPL + EC).

Table 3 illustrates the name and meaning of each variable used in the econometric model. Here, *sq* is a dummy variable representing a status quo, with 1 representing standard alternatives and 0 representing other alternatives. Amenity is a variable representing respondents' score (of 100) for amenities in the tourist destination. Guide is a variable rate on a 5-point scale for information/interpretation services in the tourist destination. Habitat is a variable in which the respondents evaluated the level of habitat quality of their most preferred tourist destination, and the input ranges from 0.00 to 1.00. Cost refers to the admission fee per person for each tourist destination.

The CL, RPL, and RPL + EC models were estimated using NLOGIT 6.0; the coefficients of the CL model were estimated through the maximum likelihood estimation method, and the coefficients of the RPL and RPL + EC models were estimated using the simulated maximum likelihood estimation. The results of the choice model estimation are summarized in Table 4. The estimated coefficient of all the variables demonstrated statistical

significance within 5%, and the sign of the coefficient was also estimated to be consistent with the expectation. First, the estimated coefficient of the *sq* variable was positive (+), so if all other conditions between the alternatives remained the same, this demonstrated that the respondents preferred the standard alternative (tourist destinations evaluated by the respondents) to a hypothetical tourist destination as an alternative. The amenities, guide, and habitat variables were all statistically significant and positive, indicating that every increasing attribute level significantly affected alternative choices. Cost, a variable representing admission fees for each tourist destination, was negative, indicating that higher admission fees negatively affected choices.

**Table 3.** Variable and definitions used in the econometric model.

| Variable | Definition |
|---|---|
| *sq* | 1 = standard alternative (tourist destinations evaluated by respondents), 0 = other alternatives |
| amenity | Satisfaction with amenities in the tourist destination (of 100 points) |
| guide | Information/interpretation service in the tourist destination (5 = very good and 1 = very poor) |
| habitat | Habitat quality in the tourist destination (0.00–1.00) |
| cost | Admission fee per adult for each tourist destination (KRW/person) |

**Table 4.** Estimation results for CL, RPL, and RPL + EC models.

| | Choice Models | | | | |
|---|---|---|---|---|---|
| **Model** | **CL** | **RPL** | | **RPL + EC** | |
| **Variable** | **Estimated Coefficient** | **Estimated Coefficient** | **Standard Deviation** | **Estimated Coefficient** | **Standard Deviation** |
| *sq* | 0.7109 8*** | 0.86359 *** | 1.67978 *** | 1.03532 *** | 1.63560 *** |
| amenity | 0.01928 *** | 0.02529 *** | 0.03389 *** | 0.02868 *** | 0.03634 *** |
| guide | 0.16162 *** | 0.19844 *** | 0.16545 ** | 0.22131 *** | 0.22694 *** |
| habitat | 1.94264 *** | 2.72644 *** | 3.33930 *** | 2.94728 *** | 3.79183 *** |
| cost | −0.00035 *** | −0.00046 *** | n/a | −0.00053 *** | n/a |
| $\sigma_2$ | - | - | - | 1.08926 *** | - |
| $\sigma_3$ | - | - | - | 0.26868 *** | - |
| LL | −17,801.6 | −16,337.9 | | −16,125.2 | |
| AIC | 35,613.3 | 32,693.9 | | 32,272.5 | |
| BIC | 35,652.3 | 32,764.1 | | 32,358.2 | |
| Pseudo R2 | 0.10 | 0.174 | | 0.185 | |

Note: *** and ** represent statistical significance at 1% and 5%, respectively.

In the RPL model, all the variables except cost are presumed to be random parameters that follow a normal distribution. The estimation results of the RPL model demonstrate that all the SDs of the random parameters were statistically significant within 5%, which supports the assumption of heterogeneity of each random parameter of preferences. Parameters such as *sq*, amenity, and habitat, the statistical significance or magnitude of the estimated coefficients did not differ much from the estimated values in the CL model. However, habitat demonstrated a difference in the magnitude of the coefficient, implying that it may affect the results of the benefit estimation for change in habitat quality in the future.

For the RPL + EC model, it was impossible to include all the ECs ($\sigma_j$); therefore, the component with the smallest variance had to be normalized. The results of the analysis demonstrated that the variance of the standard alternative was the smallest, thereby normalizing the SD ($\sigma_1$) of the error term for the standard alternative to 0. Compared to the RPL model, there

was not much difference in the coefficients of amenity, guide, and habitat, but the coefficient of *sq* demonstrated a considerable change. $\sigma_2$ and $\sigma_3$ were positive and demonstrated statistical significance within 1%, and the coefficients were 1.089 and 0.269, respectively. This implies that the variance of the standard alternative was $\pi^2/6$, whereas the variance of the hypothetical alternatives 2 and 3 had increased to $\left(\frac{\pi^2}{6}\right) + 1.089^2$ and $\left(\frac{\pi^2}{6}\right) + 0.269^2$, respectively.

Various indicators were examined to select the most suitable model. After comparing AIC, BIC, and Pseudo R2, the RPL model was preferable to the CL model. A likelihood-ratio test was conducted to compare the RPL model and the RPL + EC model. The results demonstrated that the RPL + EC model was better. Therefore, the estimation results of the RPL + EC model were used for the benefit estimation of habitat quality change. Figure 6 shows the estimated distribution of each individual's WTP using a kernel density estimator for habitat quality attributes based on the RPL + EC model.

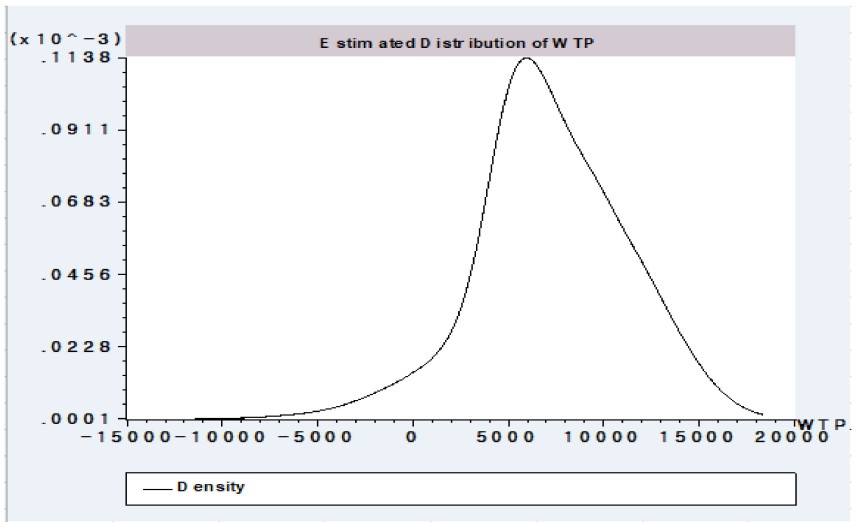

**Figure 6.** Estimated distribution of each individual's willingness-to-pay for habitat quality attribute.

### 3.2. Benefit Estimation

This study calculated the economic value of change in habitat quality for 10 years (2009–2019) in 12 major tourist destinations on Jeju and estimated the social benefits based on the number of tourists visiting each destination. Table 5 summarizes the changes in habitat quality in each destination over 10 years. The change in habitat quality varied between destinations but increased by 0.19 on average. Among the tourist destinations, Jungmun Daepo Columnar Joints demonstrated an increase (0.39), while Seogwipo Natural Recreation Forest did not demonstrate any change. The main reason for the increase in habitat quality was the fact that the forest area, which has a high degree of habitat suitability, increased by 100 km$^2$ between 2009 and 2019, and the areas covered by farmland, which has a low degree of habitat suitability, decreased by 150 km$^2$.

The social benefit associated with the change in habitat quality on Jeju was calculated by applying the compensating variation (CV) formula in Equation (3) and using the RPL+EC model's estimation results in Table 4 and the results summarized in Table 5. The benefit estimation scenarios are illustrated in Table 6.

$$CV^i = -\frac{V_1^i - V_0^i}{\beta_{cost}} = -\frac{\beta_{habitat}^i \times \left(habitat_{2019}^i - habitat_{2009}^i\right)}{\beta_{cost}}, \tag{3}$$

where *i* represents each tourist destination, and $habitat_{2019}^i$ and $habitat_{2009}^i$ represent the habitat quality of *i* tourist destinations in 2019 and 2009, respectively.

The results of the *CV* estimation according to changes in habitat quality are summarized in the second column of Table 7, and the results of social benefit estimation based on the number

of tourists who visited each destination in 2019 are illustrated in the fourth column. The results of estimating the *CV* demonstrated that, except for Seogwipo Natural Recreation Forest, which demonstrated no change in habitat quality, the *CV* ranged from a minimum of KRW 253 (Jeolmul Natural Recreation Forest) to a maximum of KRW 2302 (Bija Forest). Considering the number of visitors to tourist destinations in 2019, the social benefit of the change in habitat quality ranged from a minimum of KRW 180 million/year (Jeolmul Natural Recreation Forest) to a maximum of KRW 2.37 billion/year (Jungmun Daepo Columnar Joints). The sum of the benefits was estimated at approximately KRW 12 billion/year.

**Table 5.** Habitat quality change estimated by InVEST Habitat Quality model in 12 tourist destinations on Jeju (2009–2019).

| Tourist Destinations | 2009 | 2019 | Variation |
|---|---|---|---|
| Bija Forest | 0.45 | 0.79 | 0.34 |
| Seongsan Ilchulbong | 0.33 | 0.55 | 0.22 |
| Sangumburi Crater | 0.56 | 0.76 | 0.20 |
| Jeolmul Natural Recreation Forest | 0.79 | 0.83 | 0.04 |
| Mara Provincial Park | 0.44 | 0.57 | 0.13 |
| Sanbangsan Mountain | 0.60 | 0.69 | 0.09 |
| Yeomiji Botanic Garden | 0.33 | 0.39 | 0.06 |
| Cheonjeyeon Waterfall | 0.33 | 0.62 | 0.29 |
| Jungmun Daepo Columnar Joints | 0.13 | 0.53 | 0.39 |
| Seogwipo Natural Recreation Forest | 0.85 | 0.85 | 0.00 |
| Cheonjiyeon Waterfall | 0.20 | 0.48 | 0.27 |
| Jeongbang Waterfall | 0.00 | 0.22 | 0.22 |

**Table 6.** Benefit estimation scenario.

| Category | Detail | |
|---|---|---|
| Analysis method | Social benefit of habitat quality change in each tourist destination | |
| Scenario | Habitat quality change in each tourist destination for 10 years (2009–2019) | |
| Utility function | 2019 habitat quality $V_1^i$ | 2009 habitat quality $V_0^i$ |
| Benefit analysis model | RPL + EC model | |

**Table 7.** Results of the benefit estimation.

| Tourist Destinations | Compensating Variation (*CV*) (KRW/Person) | Social Benefit | |
|---|---|---|---|
| | | No. of Tourists in 2019 (Person) | Benefit (100 Million KRW/Year) |
| Bija Forest | 2301.7 | 799,825 | 18.4 |
| Seongsan Ilchulbong | 1180.8 | 1,707,913 | 20.2 |
| Sangumburi Crater | 1233.4 | 500,000 | 6.2 |
| Jeolmul Natural Recreation Forest | 253.0 | 729,749 | 1.8 |
| Mara Provincial Park | 661.2 | 876,843 | 5.8 |
| Sanbangsan Mountain | 452.5 | 492,880 | 2.2 |
| Yeomiji Botanic Garden | 333.4 | 1,200,000 | 4.0 |
| Cheonjeyeon Waterfall | 1596.2 | 549,464 | 8.8 |
| Jungmun Daepo Columnar Joints | 2025.0 | 1,169,852 | 23.7 |
| Seogwipo Natural Recreation Forest | 0.0 | 129,294 | 0.0 |
| Cheonjiyeon Waterfall | 1412.1 | 1,323,545 | 18.7 |
| Jeongbang Waterfall | 1321.6 | 778,717 | 10.3 |
| Total | | 1,999,825 | 120.1 |

Note: The number of tourists visiting the Sangumburi Crater was obtained from the visitor statistics collected yearly at the Sangumburi Crater (as of 6 November 2020). Source: Tourism Knowledge & Information System, "Visitor Statistics of Main Tourist Sites," applied by the author.

## 4. Discussion

As environmental issues have emerged worldwide, environmental considerations are being included in development. Since the first environmental assessment system was introduced in the United States in 1969 based on the National Environmental Policy Act(NEPA), it has been operated in various forms in many countries. While Environmental Impact Assessment(EIA) is related to the impact of development projects on the environment, Strategic Environmental Assessment(SEA) comprehensively considers environmental, economic, and social impacts when establishing national policies, plans, and programs that precede development projects. Therefore, it can be considered a systematic decision-making support tool for sustainable development.

In Korea, the legal basis was established with the enactment of the Environmental Preservation Act in 1977, and the environmental assessment system has been implemented since related regulations were announced in 1981. Since the revision of the Environmental Impact Assessment Act in 2012, it has been divided into EIA, SEA, and mini environmental impact assessment. SEA indicates an assessment of the feasibility of a plan from an environmental perspective by verifying whether the environmental impact of the plan conforms to the relevant environmental conservation guidelines and by developing and analyzing alternative ways of promoting sustainable development of national land [23].

However, this definition implies that the SEA in Korea is operated centering on the impact of environmental aspects, rather than following the original purpose of the SEA by considering all the environmental, economic, and social aspects aforementioned. In addition, the guidelines for the SEA specified in Article 4 of the same Act stipulate, EIAs, etc. shall be conducted in consideration of the social and economic impacts of environmental hazards of the plan or project on groups sensitive to exposure to environmental harmful factors, including children, senior citizens, pregnant women and nursing mothers, and low-income people [23]. Although some socio-economic factors are included, there is a lack of environmental benefits/costs.

With the acceleration of climate change, the importance of ecosystems is becoming more prominent. As environmental issues are closely related to society and the economy, it is necessary to develop an integrated evaluation system for various sectors of the Sustainable Development Goals that encompass not only the environment but also the socio-economic sector. In this respect, this study provides a basic direction of an integrated evaluation system. In other words, this study contributes to examining the possibility of supporting sustainability evaluation, that is, evaluation that considers environmental, economic, and social aspects together using environmental benefit/cost information. In addition, in order to compare and analyze the effects of land use (e.g., damage to ecosystems), it is necessary to quantify and present them objectively. In this study, environmental valuation necessary for strategic decision-making was presented systematically and quantitatively. It is proposed to include environmental benefits/costs in the evaluation of sustainable development programs. Additional case studies will be required in order to further improve the evaluation process.

In this study, our suggestions are as follows. First, the qualitative aspects of biodiversity, that is, the value of habitat quality, should be sufficiently considered when introducing development projects or policies. Through this, the implementations of a project or policy must be carefully determined by comparing the benefits of development with the potential environmental costs. As public awareness of environment increases, its costs are likely to rise further. Second, it is necessary to integrate these evaluation results into sustainability evaluation. While guidelines for environmental policies such as regulatory impact analysis(RIA), for example, specify to reflect environmental benefits and costs [24], SEA lacks relevant contents. Third, this study presents an integrated environmental and economic analysis as a systematic method for estimating the value of ecosystem habitat quality change. Because of the complexity of environmental problems, it is not easy to identify scientific pathways for impacts on humans and ecosystems. In order to evaluate the impact of various human activities on ecosystems, it is necessary to set the scope of evaluation

impact and quantify the scale of temporal and spatial impacts. It is possible to make more comprehensive decisions, when economic evaluation of quantified environmental values is rationally linked. In order for the integrated environmental and economic analysis to be utilized as a system that helps decision-making beyond individual case studies, it is essential to systematize its process.

## 5. Conclusions

While provisioning, regulating, and cultural services of ecosystems have relatively direct and short-term impacts on humans when they change, supporting services are characterized by indirect and long-term impacts on humans. In addition, unlike provisioning services, which corresponding to direct use value, supporting services are related to indirect use value that does not accompany direct consumption. For this reason, it is difficult to sufficiently account their values. In this study, we attempted to quantify the value of changes in habitat quality through an integrated analysis.

We estimated visitors' WTPs for changes in habitat quality in the 12 main tourist destinations on Jeju Island using a CE. The data were collected from the Jeju ecotourism survey conducted in 2019 to identify public awareness and preferences. We employed a pivot-style experimental design approach in developing a set of choice alternatives. The analysis was based on an assessment of the physical impact of the habitat quality of ecosystem services derived from the InVEST Habitat Quality model of Kim et al. [6].

The results of the econometric analysis indicated the presence of heterogeneity in respondents' preferences for ecotourism destinations. Specifically, we found that visitors were most sensitive to changes in habitat quality, among the other attributes. Based on the results of the estimation of CV due to the change in habitat quality, the social benefit ranged from KRW 253 to KRW 2302 per person, excluding one destination where there was no change in habitat quality. Considering the number of visitors to each tourist site in 2019, the social benefits resulting from the improvement of habitat quality in the 12 tourist sites are estimated to be approximately KRW 12 billion per year.

There have been both domestic and global discussions on the importance of biodiversity preservation. Recognizing and estimating the economic value of ecosystems can provide evidence to support these arguments and suggest directions for ecosystem management. The results of this study can imply that there is a growing public awareness of ecosystem conservation; hence, the level of habitat quality is a crucial factor in managing ecosystem services. In addition, appropriate management of ecotourism sites through efforts to enhance biodiversity benefits residents and tourists, by increasing tourists' interest and reviving local economies.

An integrated environmental and economic framework in this study enables a comprehensive analysis based on impact pathway analyses, which trace the path leading to human activity-environmental quality change receptors to evaluate the biophysical impacts and directly link them to human activities. Through a study case, our study shows how this framework could be practically utilized as an effective resource, especially when establishing the plausible economic value generated by environmental goods and services. The findings of this study provide the qualitative consideration of ecosystem services and can be used as a reference to examine the feasibility of policies or projects requiring land-use development. Regarding the management of ecosystem services, they also contribute to providing a basis for considering the importance of their benefits to society in policy decision-making. Since our findings are based on the WTPs of the public, they contribute to directly or indirectly reflecting people's opinions in policies.

**Author Contributions:** Conceptualization, H.N.K.; methodology, H.N.K.; data analysis, H.N.K.; validation, H.N.K.; writing, H.N.K.; reviewing, H.N.K.; supervision, H.N.K.; investigation, H.R.; writing, H.R.; reviewing, H.R.; editing, H.R. All authors have read and agreed to the published version of the manuscript.

**Funding:** This study is funded by the Research Project titled "An Integrated Assessment to Environmental Valuation via Impact Pathway Analysis (GP2022-09)" conducted by Korea Environment Institute (KEI).

**Data Availability Statement:** Not applicable.

**Acknowledgments:** This study was part of the Research Project titled "An Integrated Assessment to Environmental Valuation via Impact Pathway Analysis (GP2020-10)" conducted by Korea Environment Institute (KEI).

**Conflicts of Interest:** The authors declare no conflict of interest.

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
