# Peer review of "Estimating the Economic Value of Change in Ecosystem Habitat Quality in South Korea Using an Integrated Environmental and Economic Analysis"

_land, doi:10.3390/land11122249_

Round 1

Reviewer 1 Report (Previous Reviewer 3)

The revised manuscript can be accepted in present form.

Author Response

Reviewer 2 Report (Previous Reviewer 2)

The revised manuscript is well-improved and can be accepted for publication.

Author Response

Reviewer 3 Report (Previous Reviewer 1)

Manuscript titled Estimating the Economic Value of Change in Ecosystem Habitat Quality in South Korea Using an Integrated Environmental and Economic Analysis.

After the authors did their best to revise it, I am very pleased to see that this manuscript has changed for the better compared to the first draft. I suggest that the discussion section of the manuscript could be explored in more depth. While the conclusion section of the manuscript could be more concise.

Author Response

This manuscript is a resubmission of an earlier submission. The following is a list of the peer review reports and author responses from that submission.

Round 1

Reviewer 1 Report

Manuscript titled Estimating the Economic Value of Change in Ecosystem Habitat Quality in South Korea Using an Integrated Environmental and Economic Analysis. In this study, the economic value and benefits of habitat quality changes on Jeju Island were estimated through an integrated environmental and economic analysis linking the unit values of ecosystem services to the physical habitat quality measures predicted by the InVEST Habitat Quality Model. A choice experiment survey was conducted to estimate the willingness-to-pay for the changes in habitat quality at 12 tourist sites using a hybrid econometric model. This manuscript has the following problems in general.

1. Most obviously, the tables and structural diagrams have not been adjusted according to the requirements of the journal.

2. It is suggested that the first part should be combined with the second part. In addition, the second part of the literature review needs to be improved in terms of academic quality and the textual presentation needs to be condensed.

3. It is suggested that the third part of the manuscript, the survey part, be integrated into the fourth part, the material part of the empirical study, for a coherent analysis.

4. The conclusion and discussion of the manuscript need to be studied in depth, and it is suggested that the manuscript be adjusted according to the basic pattern of Land to make it more scientific.

Author Response

Thank you for your comments. Please check the attached file.

Reviewer 2 Report

Please check the following comments:

Please explain the contribution of the study a little more, e.g., how this is going to solve a socioeconomic/environmental problem, how this will help to achieve any targets of the Sustainable Development Goals, etc.

Add the sources for figures 4 and 5.

The manuscript lacks sufficient discussions based on the results. Please make a separate section on 'Discussion' and explain your findings. Compare the findings with existing studies.

What you have written in the conclusion is basically a summary of the study. Clearly mention how the message of this study can be applied to broader aspects. 

Author Response

(The authors gave the same response as above.)

Reviewer 3 Report

The literature review is incomplete. Please refer to many more recent scientific publications that are thematically related to your manuscript. It is difficult to make a review of an article, the authors of which have not made sure whether similar studies have been carried out so far or which results from other similar analyzes carried out by other researchers.

In the chapter Conclusions, the authors write: "The data were collected from the Jeju ecotourism survey conducted in 2019 to identify public awareness and preferences. We employed a pivot-style experimental design approach in developing a set of choice alternatives." Are the test results up-to-date? What has changed in the last three years? Has the coronavirus pandemic changed anything?

In the chapter Conclusions, please emphasize clearly the importance of the research carried out and the possible practical application of the analyzes carried out in the manuscript. Please note that perhaps an extended literature review will have an impact on the chapter Conclusions.

Author Response

(The authors gave the same response as above.)
